# Tocopherol more bioavailable than tocopheryl-acetate as a source of vitamin E for broilers

Theo A. T. G. van Kempen[1,2]*, Samuel Benítez Puñal[2], Jet Huijser[2], Stefaan De Smet[3]

1 Trouw Nutrition, Boxmeer, Netherlands, 2 Department of Animal Science, North Carolina State University, Raleigh, NC, United States of America, 3 Laboratory of Animal Nutrition and Animal Product Quality, Faculty of Bioscience Engineering, Ghent University, Ghent, Belgium

* theovankempen@yahoo.com

**Data Availability Statement:** All relevant data are within the paper and its Supporting Information files.

**Funding:** 3 of the authors are employed by the funding company, and the research was carried out

## Abstract

Vitamin E is typically supplied in the form of tocopheryl-acetate (**T-Ac**) since tocopherol (**T**) has stability issues. Tocopheryl-acetate, however, must be hydrolyzed in the intestines before it can be absorbed, a step that is purportedly rate-limiting for its bioavailability. The objective of this study was to compare the efficiency of absorption of T-Ac and T in broilers. In addition, two test procedures were evaluated in which animals received the test substances for either 2 or 4 days only. Animals were adapted to diets without supplemental vitamin E (feedstuffs contributed 14±1 ppm natural vitamin E (RRR-tocopherol)) till the age of 25 d (individual housing) or 28 d (group housing). Subsequently, they were fed T-Ac at 80, 53, 36, 24, or 16 ppm or T at 80, 40, 20, 10, or 5 ppm for a period of 4 d (**4-di**) or 2 d (**2-dg**), after which serum and liver were collected for analysis of vitamin E. Measured feed vitamin E levels were used for the data analysis; the recovery of T-Ac was 85%, and that of T was 39%. Both test procedures (2 or 4 days) yielded good quality data. Based on linear regression analysis, the relative efficiency with which T-Ac raised tissue levels as compared to T was 0.24 (2-dg) to 0.37 (4-di), with liver and serum yielding similar results. Analysis using more complex dose response models imply that the hydrolysis of T-Ac was strongly dose-dependent and that it could be saturated at doses above approximately 50 ppm in animals only briefly fed T-Ac; for T there was no evidence of saturation. These data imply that T, provided that stable forms can be developed, has the potential to be much more efficient at providing vitamin E to the animal, and on top, can yield much higher tissue levels, than T-Ac.

## Introduction

Vitamin E is a lipid-soluble antioxidant in vivo and is routinely supplemented into livestock diets. Dietary vitamin E is considered to positively affect meat quality, immune response, and performance of broiler chickens [1]. The form used most for supplemented vitamin E is all-rac (racemic blend) α-tocopheryl acetate (**T-Ac**), whereby alpha-tocopherol (**T**) is acetylated for stability during storage and handling. This step, however, negatively affects the biological value

in house (but supported by sample analysis by UGent). The trial itself was done on request of the employer with the aim as outlined in the paper, and the analyses were carried out to test that aim. The trial was designed and analyzed by TvK; JH wrote the protocol, SBP was responsible for the animal trial. SDS was responsible for some of the vit. E analyses and provided input on trial design and interpretation.

**Competing interests:** van Kempen, Punal, and Huijser are employed by the company that funded the work. This does not alter our adherence to PLOS ONE policies on sharing data and materials.

**Abbreviations:** 2-dg, 2 days group housed; 4-di, 4 days individually housed; T, α-Tocopherol (all-racemic); T-Ac, α-Tocopheryl-acetate (all-racemic).

of T-Ac relative to T. Desmarchelier et al. [2] showed that in vitro T-Ac is only partially hydrolyzed to T (less than 30% in a cereal matrix). Thus, the acetylation of T into T-Ac impedes bioavailability. Indeed, previously, we showed that in vivo bioavailability of T-Ac was only 5.4% in swine [3] when fed at 75 ppm, implying that there is ample room for improvement. Tocopherol in the free form suffers from stability problems. If stable forms could be identified, however, then bioavailability advantages can be expected. The objectives of this experiment were two-fold. 1) Develop a simple yet robust test model for assessing relative bioavailability of vitamin E forms. For this, birds were fed the experimental diets for 2 or 4 days either housed in groups or individually and responses in liver and plasma were collected for analysis. Two different but both short feeding periods were used with the aim to be in a linear response phase (no tissue saturation) but at the same time with the aim was to see clear tissue responses. 2) To compare the bioavailability of T with the industry-standard T-Ac in broilers. For this, tissue T levels from broilers that consumed diets supplemented with either graded levels of T-Ac or T were compared.

## Materials and methods

### Animal care

Animal care procedures were in accordance with the Spanish law Real Decreto RD 53/2013 of the 1st of February which establishes basic rules applied for protecting animals used for research, and other scientific matters like teaching. The regulations conform to the European Directive 2010/63/EU for the Protection of Vertebrate Animals Used for Experimental and Other Scientific Purposes (Brussels, European Union) and achieve the standard of care required by the US Department of Health and Human Services' Guide for the Care and Use of Laboratory Animals.

### Test materials

All-racemic tocopheryl-acetate (T-Ac: 1 mg = 1 IU) was premix-grade commercial material on a silica carrier (50%) and all-racemic tocopherol (T: 1 mg = 0.91 IU) was purchased from Sigma Aldrich (Zwijndrecht, NL). Instead of IU, levels are listed as ppm or mg to avoid confusion as both forms were used. The term vitamin E is used in this paper as a generic descriptor encompassing both forms.

### Animals

Healthy male broilers ($n$ = 800, Ross 308) were supplied by a local hatchery. In the hatchery, they were vaccinated against Marek's disease, infectious bronchitis, coccidiosis, and Gumboro. Two tests were carried out with these birds. All of them were raised in group pens till 11 days of age (20 pens with 40 birds each; the first 3 days lights were on continuously, and for the remainder of the trail lights were on from 5:00–23:00). At that point, 160 birds were removed (8 per pen) and placed in individual cages in another barn. The remaining birds were left in their pens. All birds were adapted for the entire period on a diet without supplemental vitamin E and formulated using feedstuffs that were expected to be low in vitamin E (the measured vitamin E content in the basal diets was 14±1 ppm). Otherwise, diets were formulated in line with commercial practices (Tables 1 and 2).

Birds were weighed upon receipt, and subsequently on days 11 and at the beginning and end of the test period; for group-housed birds on day 28 and 30, and for individually housed birds on days 25 and 29. Feed intake was recorded for each period on a pen basis.

**Table 1. Feedstuff composition (g/kg) of the adaptation diets (starter and grower phase) and experimental diet.**

| | Starter | Grower | Experimental |
|---|---|---|---|
| Wheat | 100 | 0 | 0 |
| Maize | 376 | 496 | 530 |
| Soybean meal 48% CP | 278 | 192 | 175 |
| Barley | 100 | 150 | 149 |
| Soy oil | 54 | 48 | 53 |
| Soycomil® | 51 | 82 | 63 |
| Calcium carbonate fine | 15.1 | 11.5 | 11.1 |
| Monocalcium phosphate | 9.9 | 4.3 | 1.8 |
| Na bicarbonate | 2.5 | 2.4 | 2.4 |
| Salt (NaCl) | 1.8 | 1.7 | 1.7 |
| L-Lysine HCl 98% | 1.6 | 1.8 | 2.1 |
| Dl-Methionine 99% | 2.4 | 2.4 | 2.3 |
| L-Threonine 98% | 0.1 | 0.3 | 0.5 |
| L-Valine 96.5% | 0.0 | 0.0 | 0.1 |
| Xylanase & ß-glucanase | 1.0 | 1.0 | 1.0 |
| Phytase | 1.0 | 1.0 | 1.0 |
| Premix[1] | 5.0 | 5.0 | 5.0 |

[1]The premix for each of the phases provided, on a final feed basis: Zn (80 ppm), Cu (15 ppm), Mn (90 ppm), I (1.1 ppm), Se (0.25 ppm), Fe (65 ppm), thiamin (2 ppm), pyridoxine HCl (4 ppm), niacin (40 ppm), pantothenic acid (10 ppm), folic acid (1 ppm), choline (300 ppm), biotin (0.15 ppm), vit. B12 (0.025 ppm), Vit. A (10000 IU), vit. D3 (2500 IU), and Vit. K (2 ppm).

## Dietary treatments

Birds were assigned to treatment using a randomized complete block design (with location in the facility as the blocking factor) to either T or T-Ac supplied at 5 different levels with either 16 individual birds or 2 pens with 32 birds each for each dose x treatment combination. As an

**Table 2. Nutrient composition (g/kg, except where noted) of the adaptation and experimental diets.**

| | Starter | Grower | Experimental |
|---|---|---|---|
| Dry matter | 891 | 887 | 886 |
| Crude protein | 215 | 196 | 179 |
| Crude fiber | 29 | 29 | 28 |
| Ash | 61 | 50 | 45 |
| Oil (ether extract) | 77 | 74 | 80 |
| NSP | 169 | 160 | 156 |
| Starch (Ewers) | 337 | 385 | 406 |
| NDF | 104 | 101 | 101 |
| ADF | 41 | 39 | 38 |
| ME broiler (kcal/kg) | 2850 | 2925 | 3000 |
| Digestible Lysine | 11.5 | 10.6 | 9.8 |
| Digestible Methionine | 5.2 | 5.0 | 4.7 |
| Digestible Methionine+Cysteine | 8.1 | 7.6 | 7.2 |
| Digestible Threonine | 6.9 | 6.5 | 6.1 |
| Digestible Tryptophan | 2.2 | 1.9 | 1.7 |
| Ca | 10.0 | 7.5 | 6.9 |
| Available P poultry | 4.8 | 3.5 | 2.9 |

upper limit vit. E level, 80 ppm was chosen for both treatments, in line with a commercially relevant upper limit. It was anticipated that T was more efficient at raising serum levels. Consequently, doses for each titration step were reduced by a factor 2 for T (thus, 80, 40, 20, 10, and 5.0 ppm), and by a factor of only 1.5 for T-Ac (thus, 80, 53, 36, 24, and 16 ppm; this range corresponds well with the range used in commercial practice). In order to homogeneously blend the test products into the final feed, test material for each experimental diet was first blended with basal diet for a total 2 kg blend; this 2 kg blend was mixed into the feed at an inclusion rate of 0.5%.

The experimental diet was based on a single batch of basal feed to which the test components were added and was manufactured one month prior to use and stored at approximately 13 ± 5°C. To correct for stability issues, diets were analyzed for vitamin E around the time of the actual animal feeding. The procedure used was derived from NEN-EN 12822:2014 [4].

## Sample collection

At the age of 25d (individually housed; identified as **4-di**) or 28d (group housed, identified as **2-dg**), birds were switched to the experimental diets. Four (4-di) or 2 (2-dg) days later, at the age of 29 or 30 days, respectively, 10 typical birds from the individual housing, and 5 typical birds per pen from the group housing for a total of 10 birds per treatment x dose for each of the housing systems were selected. The difference in schedule between the two groups was a result of labor availability, but on top the procedures were experimental and it was not clear how many days of feeding would yield clear responses hence different time frames were used. From these birds, serum and liver were collected following decapitation for analysis of vitamin E levels. Samples were flash frozen and stored at -80°C until analysis. Serum samples were analyzed using a kit (Chromsystems, Munich, Germany). Liver samples (2-dg only) were analyzed as described before [3].

## Data analysis

For both experimental designs, birds were treated as the experimental unit. Tissue concentrations were analyzed with various dose response models (described in results and discussion). Using the modeled data, an efficiency ratio was also calculated. For this the increase in tissue concentrations was divided by the increase in feed concentration for each form, and the ratio of these values for T-Ac/T was calculated. This ratio illustrates how efficient T-Ac increases tissue vitamin E levels as compared to T.

As the modeled data could not be compared statistically (see results and discussion), an analysis of variance was performed with treatment, block, and body weight at the start as of the supplementation period as covariable. Non-design parameters with P > 0.20 were removed before the final analysis (both block and body weight). The data were also modeled using linear models (using a common intercept for the 2 treatment groups). Data were analyzed using SAS JMP Pro 15 (Cary, NC, USA).

## Results and discussion

Despite being raised on a diet devoid of supplemental vitamin E up to day 25 or 28, birds performed well. For the birds in individual cages, body weight on 29 days averaged 1788 g while the Ross standard is 1649 g. For the birds raised in group the day 30 body weight averaged 1952 g while the Ross standard is 1746 g.

During the test period (Table 3), the average daily gain was 111.0 ± 17.1 g/d for individual, and 123.9±5.3 g/d for group housing (pen basis), and the feed conversion ratio 1.579 ± 0.150 g/d for individual, and 1.514±0.045 g/d for group housing. As the trial was not designed for

**Table 3. Performance data during the test period (DWG = daily weight gain, g/d; DFI = daily feed intake, g/d; FCR = food conversion ratio).**

|  | 4-di | | | 2-dg | | |
|---|---|---|---|---|---|---|
|  | **DWG25-29** | **DFI25-29** | **FCR25-29** | **DWG28-30** | **estDFI28-30** | **FCR28-30** |
| T-Ac 16 ppm | 114.0 | 175.1 | 1.55 | 129.9 | 189.9 | 1.46 |
| T-Ac 24 ppm | 104.8 | 162.9 | 1.58 | 120.1 | 178.7 | 1.49 |
| T-Ac 36 ppm | 114.7 | 173.4 | 1.52 | 122.4 | 184.4 | 1.51 |
| T-Ac 53 ppm | 108.3 | 174.8 | 1.62 | 122.0 | 191.1 | 1.57 |
| T-Ac 80 ppm | 110.2 | 171.9 | 1.58 | 119.7 | 186.8 | 1.56 |
| T 5 ppm | 97.9 | 159.9 | 1.65 | 124.0 | 190.7 | 1.54 |
| T 10 ppm | 112.6 | 173.2 | 1.56 | 130.9 | 191.1 | 1.46 |
| T 20 ppm | 113.4 | 179.5 | 1.60 | 119.7 | 184.7 | 1.55 |
| T 40 ppm | 116.6 | 179.5 | 1.54 | 127.6 | 191.3 | 1.50 |
| T 80 ppm | 117.5 | 186.7 | 1.59 | 123.1 | 186.3 | 1.51 |
| SD | 17.1 | 20.4 | 0.15 | 3.4 | 3.1 | 0.03 |

studying performance effects (number of observations, length of test period), conclusions based on performance should not be drawn.

Feeds were analyzed for actual vitamin E content around the time the experimental diets were tested in the animals. Diets had a background vitamin E level of 14±1 ppm. For added T-Ac, the recovery was 85%, and for added T the recovery was 39% ($R^2$ between formulated and analyzed for T-Ac was 0.99 and for T 0.94; Fig 1). Added vitamin E contents, corrected for these recovery figures, were used for the presentation and interpretation of the results.

Serum vitamin E levels were well in line with expectations. The intercept of the modeled data showed values around 20 µM (Table 4), while the highest value observed (per group) was 51.9±7.2 µM for T fed at 31.4 ppm for 4 days. These values correspond well with the range observed by Jensen et al. [5]. For the group-housed birds both serum and liver vitamins were analyzed (Fig 2). Liver values were in line with those observed by Bottje et al. [6]. A regression

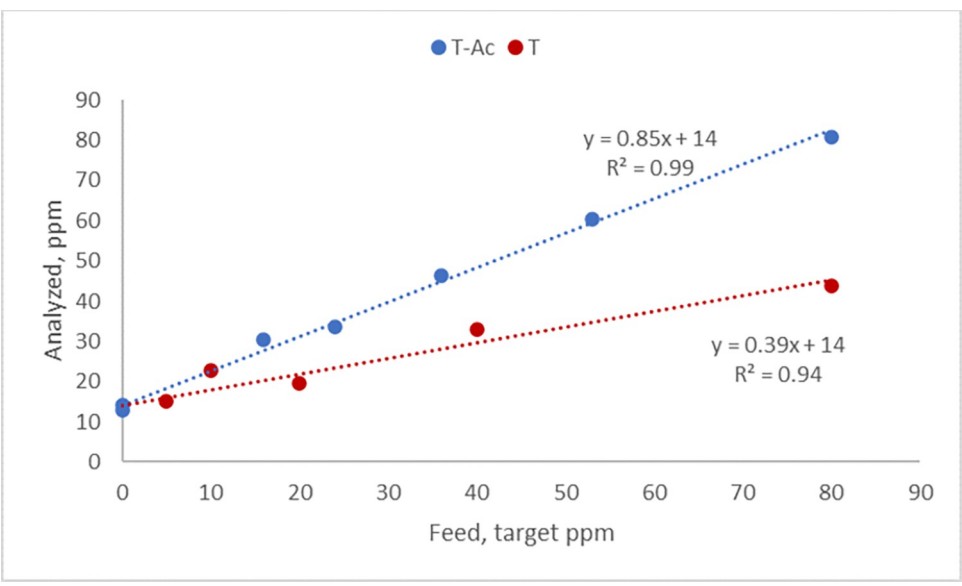

**Fig 1. Analyzed vs. formulated vitamin E levels in the experimental diets.**

**Table 4. Regression coefficients and efficiency ratio using linear models (serum data in μM, liver data in μg/g, dietary levels in ppm; T-Ac = tocopheryl-acetate, T = tocopherol).**

| Housing | 2-d serum group | 4-d serum individual | 2-d liver group |
|---|---|---|---|
| Constant | 19.4±3.5 | 24.3±4.0 | 12.6±4.0 |
| slope T-Ac | 0.17±0.10 | 0.32±0.12 | 0.25±0.13 |
| slope T | 0.72±0.24 | 0.85±0.28 | 0.88±0.27 |
| Efficiency ratio | 0.24 | 0.37 | 0.28 |
| P slope T-Ac vs. T | 0.00 | 0.00 | 0.00 |
| $R^2$ | 0.46 | 0.52 | 0.57 |

between both showed that they correlated well ($R^2 = 0.77$), and that this correlation was not affected by treatment (P>0.20).

The original intent was to model the data using an exponential decay model with the constant (intercept) being the same for both treatment groups. This model, however, did not describe the data adequately. For T, the data could be described using a linear model, while for T-Ac the data resembled an exponential decay model but with a lag phase. The use of two separate models, however, renders the modeled data incomparable from a statistics perspective. Consequently, we had to improvise in how to analyze the data.

One such improvisation was to analyze the data using linear models using a common intercept only (Table 4). Mechanistically a linear model is not ideal, but to answer the primary question of this study this model does provide usable answers. T was absorbed much more efficient than T-Ac for both the 2-dg and for the 4-di serum data, as well as for the 2-dg liver data. The ratio of the slopes indicates an efficiency of 0.24 to 0.37 of T-Ac vs. T in increasing tissue T levels. Interestingly, the slope for T-Ac was nearly twice for 4-di than for 2-dg birds; in contrast, the slopes for T were practically identical for 4-di and 2-dg birds. It is not clear why 4-di feeding resulted in a steeper slope than 2-dg feeding. Adaptation as a result of a longer feeding period is arguably a poor argument as the enzyme responsible for this hydrolysis, carboxyl ester hydrolysis, has a generic role in fat digestion while tocopheryl ester hydrolysis is likely only a fringe activity [7]. Chung et al. [8] obtained an efficiency ratio of 0.41 of T-Ac vs. T in piglets fed test diets over a 20-d period, well in line with our results. In contrast, Burton et al. [9] reported an efficiency ratio close to 1; however, these authors did not correct for the instability of T, but neither did Chung et al. [8].

Analyzing the data assuming that each treatment is independent showed that there are strong differences (Table 5). E.g., T at the highest dose (31 ppm) yielded serum responses that were significantly higher than T-Ac at 30 ppm (P<0.05). The highest dose of T-Ac, 68 ppm, yielded numerically and statistically the same response as 45 ppm T-Ac implying that the response was saturated. These LSMeans values are also incorporated in the graphs in Fig 3.

For discussion purposes, data were modeled with an exponential decay model incorporating a lag factor which described the T-Ac data well:

If dose < lag, Constant,

If dose > lag, $A^*(1-exp^{(-K^*(dose-lag))})$ + Constant

Constant = intercept at 0 dose, A = maximal response at dose infinity, K = rate constant, lag = maximal dose at which no increase in tissue level is observed.

For the T group, the model also described the response well, but the model wasn't stable as K approached 0 and the plateau fell far outside of the test range; practically a linear model would have described the data better. Consequently, data (Fig 3) are reported for discussion purposes only and without error measures nor statistics.

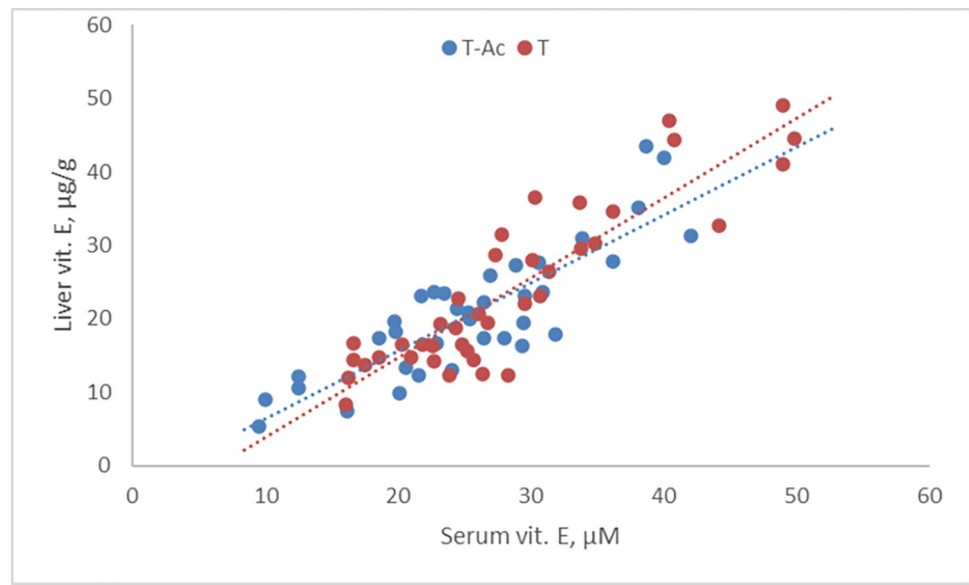

**Fig 2. Comparison of serum and liver vitamin E levels for the group-housed birds.** Control birds received tocopherol-acetate (T-Ac), and test birds received tocopherol (T).

The intercept for both groups of animals, or the serum vit. E levels assuming a dose of zero supplemental vitamin E was approximately 20 μM. In the birds fed T-Ac for 2 days only, a lag phase in serum and liver vitamin E response was observed; doses of 21 (serum) or 28 (liver) ppm were needed before an increase in tissue level was observed. For birds fed T-Ac for 4 days, this lag was only 8 ppm. Per above, adaptation could be an explanation (as proposed by Hui and Howles [10]), but given the generic role of carboxyl ester hydrolase we deem this unlikely unless vitamin E activates the transcription of the gene encoding this hydrolase.

Adding additional T-Ac to the diet resulted in an exponential decay response; this implies that the higher the dose, the lower the efficiency with which vit. E increases serum levels. Based on the modeled results, the theoretical ceiling in response (parameter A in the model) was a delta serum vit. E of 9 μM (2-dg administration) or 20 μM (4-di administration) and for liver 12 μg/g for T-Ac while 90% of the maximal response was achieved at a dose of 45, 58, or

**Table 5. Serum (μM) and liver (μg/g) vitamin E levels analyzed using traditional analysis of variance using a Tukey comparison (T-Ac = tocopheryl-acetate, T = tocopherol).**

| Vit. E | Dose, ppm | d-2g serum | | d-4i serum | | d-2g liver | |
|--------|-----------|------------|------|------------|------|------------|------|
| T-Ac | 13.4 | 22.0 | bcd | 28.7 | cd | 15.2 | d |
| T-Ac | 20.2 | 20.6 | cd | 33.6 | bcd | 14.3 | d |
| T-Ac | 30.3 | 25.7 | bcd | 35.1 | bcd | 20.2 | bcd |
| T-Ac | 45.4 | 30.4 | b | 41.5 | ab | 27.2 | b |
| T-Ac | 68.1 | 29.3 | bc | 42.5 | ab | 27.6 | abc |
| T | 2.0 | 19.2 | d | 24.4 | d | 15.2 | cd |
| T | 3.9 | 24.8 | bcd | 24.9 | cd | 16.0 | cd |
| T | 7.8 | 24.5 | bcd | 32.4 | bcd | 19.2 | bcd |
| T | 15.7 | 31.3 | b | 36.1 | bc | 28.4 | ab |
| T | 31.4 | 43.6 | a | 51.9 | a | 39.1 | a |
| SD | | 6.5 | | 7.2 | | 7.2 | |

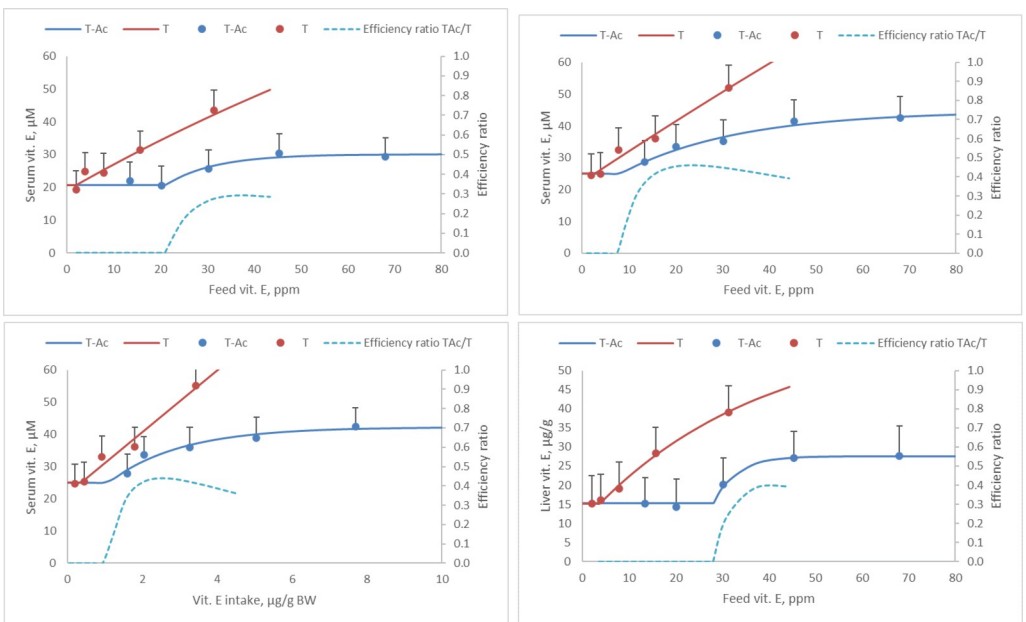

**Fig 3.** Tissue vitamin E in relation to the dietary level (Fig 3A, 3B, and 3D) or intake (Fig 3C) of tocopherol (T) or tocopheryl-acetate (T-Ac), as well as the efficiency ratio by which each dietary vitamin E source raises serum levels. Graphs show both the modeled results (lines) as well as the LSmeans obtained using ANOVA: A: serum data from group-housed birds fed the test diets for 2 days (2-dg, $R^2$ = 0.48). B: serum data from individually housed birds fed the test diets for 4 days (4-di, $R^2$ = 0.54). C: serum data from individually housed birds fed the test diets for 4 days; analysis using actual vitamin E intakes (4-di, $R^2$ = 0.60). D: liver data from the group-housed birds fed the test diets for 2 days (2-dg, $R^2$ = 0.60).

39 ppm, respectively (in line with the inclusion rates commonly used in commercial diets). These results imply that the hydrolysis of T-Ac into T is not a fixed value, but saturable. If indeed the esterase activity can be induced, then higher doses of T-Ac may still be effective at raising tissue levels in birds adapted longer to T-Ac. It should be noted that others obtained quite different dose response curves. E.g., Guo et al. [11] tested T-Ac up to 100 ppm and failed to see any saturation in plasma levels, but his absolute tissue values do not correspond with ours. Jensen et al. [5] tested T-Ac up to 200 ppm: both in liver and plasma effectively a linear increase was observed up to 150 ppm. Above 150 ppm liver levels continued to increase; plasma levels appeared to be leveling off. In contrast, Li et al. [12] tested T-Ac up to 200 ppm; his data show that, in muscle, practically speaking, levels plateaued above 100 ppm.

For T, however, an effectively linear response was observed for the serum data (both the lag and the K value obtained in the models were practically zero). The theoretical delta serum vit E was 117 (2-dg) or 395 (4-di) μM and for liver vit E 46 μg/g, respectively, while 90% of the maximal response was achieved at a dose of 335 (2-dg), 942 (4-di), or 89 (liver) ppm. These values are well beyond the range tested and hence theoretical, but nevertheless these data show that T-Ac has a much lower efficiency in raising serum vitamin E, and this efficiency diminishes quickly with increasing doses, as compared to T.

For the individually housed birds, vitamin E intake was calculated and used as input for regression analysis. This yielded practically the same response curve, but with smaller error terms for the model parameters (Fig 3C). Using individually housed animals a trial with less experimental units would thus suffice.

The calculated efficiency ratio using modeled data (Fig 3) shows that this factor is dependent on the dose; at moderate inclusions T-Ac failed to raise tissue levels while T did, implying

an efficiency of 0. At intermediate levels of supplementation (20–40 ppm) a peak efficiency ratio around 0.30–0.45 was observed. This value is well in line with the values reported by Desmarchelier et al. [2]. However, as this value is strongly affected by dose, care should be taken to extrapolate this efficiency to all inclusion levels. At higher doses, extrapolating our data suggests that this efficiency ratio drops again as the hydrolysis of T-Ac seems to be saturated while the absorption of T is not. As high levels of T were not tested, though, we can not draw a firm conclusion on this.

No efforts were made in this study to assess the impact of vitamin E on for example oxidative stress. As such, no recommendations can be made on optimal inclusion.

In conclusion, supplying test diets for only 2 or 4 days provided high quality dose-response data with no evidence for tissue saturation and still clear increases as a consequence of increasing doses. Housing animals individually would allow for less experimental units. As to bioavailability, following short term exposure (2 to 4 days), T was utilized more efficiently than T-Ac for raising tissue vit. E, and on top the efficiency of T-Ac to raise tissue levels appeared to plateau at much lower concentrations than T. Combined, stable sources of tocopherol have the potential to be much more effective sources of vitamin E for poultry with which much higher serum concentrations can be achieved.

## Supporting information

**S1 Data.**
(XLSX)

## Acknowledgments

We thank Els Vossen, Ghent University and Maastricht University for the analyses of samples and the staff at the Poultry Research Centre, Spain, for carrying out the animal experiments.

## Author Contributions

**Conceptualization:** Theo A. T. G. van Kempen, Jet Huijser, Stefaan De Smet.

**Data curation:** Theo A. T. G. van Kempen, Stefaan De Smet.

**Formal analysis:** Theo A. T. G. van Kempen.

**Funding acquisition:** Theo A. T. G. van Kempen.

**Investigation:** Theo A. T. G. van Kempen, Samuel Benítez Puñal, Jet Huijser, Stefaan De Smet.

**Methodology:** Theo A. T. G. van Kempen, Samuel Benítez Puñal, Stefaan De Smet.

**Project administration:** Theo A. T. G. van Kempen, Jet Huijser.

**Resources:** Theo A. T. G. van Kempen.

**Supervision:** Jet Huijser.

**Validation:** Theo A. T. G. van Kempen.

**Visualization:** Theo A. T. G. van Kempen.

**Writing – original draft:** Theo A. T. G. van Kempen, Stefaan De Smet.

**Writing – review & editing:** Theo A. T. G. van Kempen, Samuel Benítez Puñal, Jet Huijser, Stefaan De Smet.

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
