## [Decision Letter · Decision Letter 0]

18 Feb 2022

PONE-D-22-00873Tocopherol more bioavailable than tocopheryl-acetate as a source of vitamin E for broilers.PLOS ONE

Dear Dr. Theo van Kempen,

Thank you for submitting your manuscript to PLOS ONE. After careful consideration, we feel that it has merit but does not fully meet PLOS ONE’s publication criteria as it currently stands. Therefore, we invite you to submit a revised version of the manuscript that addresses the points raised during the review process.

We look forward to receiving your revised manuscript.

Kind regards,

Ewa Tomaszewska, DVM Ph.D

Academic Editor

PLOS ONE

Journal Requirements:

 [The work was funded by the employer of several of the authors]. 

We note that one or more of the authors is affiliated with the funding organization, indicating the funder may have had some role in the design, data collection, analysis or preparation of your manuscript for publication; in other words, the funder played an indirect role through the participation of the co-authors. If the funding organization did not play a role in the study design, data collection and analysis, decision to publish, or preparation of the manuscript and only provided financial support in the form of authors' salaries and/or research materials, please do the following:

a) Review your statements relating to the author contributions, and ensure you have specifically and accurately indicated the role(s) that these authors had in your study. These amendments should be made in the online form.

b) Confirm in your cover letter that you agree with the following statement, and we will change the online submission form on your behalf: 

“The funder provided support in the form of salaries for authors [insert relevant initials], but did not have any additional role in the study design, data collection and analysis, decision to publish, or preparation of the manuscript. The specific roles of these authors are articulated in the ‘author contributions’ section.

[van Kempen, Punal, and Huijser are employed by the company that funded the work]. 

Additional Editor Comments:

Authors shoud explain why such period of the study was chosen; next - the composition of premixes shoud be given; how the studied doses were chosen, based on what; the production performance should be given.

Reviewers' comments:

Reviewer's Responses to Questions

**Comments to the Author**

1. Is the manuscript technically sound, and do the data support the conclusions?

Reviewer #1: Yes

2. Has the statistical analysis been performed appropriately and rigorously? 

Reviewer #1: I Don't Know

3. Have the authors made all data underlying the findings in their manuscript fully available?

Reviewer #1: Yes

4. Is the manuscript presented in an intelligible fashion and written in standard English?

Reviewer #1: Yes

5. Review Comments to the Author

Reviewer #1: Comments to the Authors of manuscript number: PONE-D-22-00873 entitled „Tocopherol more bioavailable than tocopheryl-acetate as a source of vitamin E for broilers”.

Major concerns:

Line 24-25: In addition, two test procedures were evaluated in which animals received the test substances for either 2 or 4 days only. Why was this done? Please explain.

Linia 99-100: what about the content of vitamins in the starter and grower premixes? Was it the same?

Line 112: “…supplied at 5 different levels…” – what levels exactly? The experimental setup should be presented in a table, as it is illegible in the present form. The choice and level of the factor should be explicit and clear. What was the basis for the choice of these levels?

Line 117-118: “…which the test components (contained in a premix)....” - In the composition of the premixes, the amounts or levels of the tested additives should be specified below the table or separately in another table.

Line 172: “Serum vitamin E levels were well in line with expectations.” Why was it done if the result was known in advance?

Line 284-285: “No efforts were made in this study to assess the impact of vitamin E on for example

oxidative stress. As such, no recommendations can be made on optimal inclusion.” It is a pity. There are also no production performance results indicated by the research, which would increase the value of the study.

In the discussion, the authors focus on comparison of their research results with studies conducted by other authors. There is no precise interpretation of the results, which is necessary to regard the publication as valuable. Even when the authors try to explain the results of their research, they actually repeat the information contained in the literature they refer to. There is no broader view of the problem. Therefore, I believe that this section is insufficient and needs improvement.

6. PLOS authors have the option to publish the peer review history of their article (what does this mean?). If published, this will include your full peer review and any attached files.

Reviewer #1: No

---

## [Author Response · Author response to Decision Letter 0]

5 Apr 2022

Comments are provided in a word file submitted with the manuscript.

---

## [Decision Letter · Decision Letter 1]

11 May 2022

Tocopherol more bioavailable than tocopheryl-acetate as a source of vitamin E for broilers.

PONE-D-22-00873R1

Dear Dr. Theo van Kempen,

We’re pleased to inform you that your manuscript has been judged scientifically suitable for publication and will be formally accepted for publication once it meets all outstanding technical requirements.

Kind regards,

Ewa Tomaszewska, DVM Ph.D

Academic Editor

PLOS ONE

Reviewers' comments:

Reviewer's Responses to Questions

**Comments to the Author**

1. If the authors have adequately addressed your comments raised in a previous round of review and you feel that this manuscript is now acceptable for publication, you may indicate that here to bypass the “Comments to the Author” section, enter your conflict of interest statement in the “Confidential to Editor” section, and submit your "Accept" recommendation.

Reviewer #1: (No Response)

2. Is the manuscript technically sound, and do the data support the conclusions?

Reviewer #1: Yes

3. Has the statistical analysis been performed appropriately and rigorously? 

Reviewer #1: I Don't Know

4. Have the authors made all data underlying the findings in their manuscript fully available?

Reviewer #1: Yes

5. Is the manuscript presented in an intelligible fashion and written in standard English?

Reviewer #1: Yes

6. Review Comments to the Author

Reviewer #1: (No Response)

7. PLOS authors have the option to publish the peer review history of their article (what does this mean?). If published, this will include your full peer review and any attached files.

Reviewer #1: No

---

## [Editor Report · Acceptance letter]

16 May 2022

PONE-D-22-00873R1 

Tocopherol more bioavailable than tocopheryl-acetate as a source of vitamin E for broilers. 

Dear Dr. van Kempen:

I'm pleased to inform you that your manuscript has been deemed suitable for publication in PLOS ONE. Congratulations! Your manuscript is now with our production department. 

Kind regards, 

on behalf of

Professor Ewa Tomaszewska 

Academic Editor

PLOS ONE